# Sensitive detection of rare disease-associated cell subsets via representation learning

Eirini Arvaniti[1,2,3] & Manfred Claassen[1,2]

Rare cell populations play a pivotal role in the initiation and progression of diseases such as cancer. However, the identification of such subpopulations remains a difficult task. This work describes CellCnn, a representation learning approach to detect rare cell subsets associated with disease using high-dimensional single-cell measurements. Using CellCnn, we identify paracrine signalling-, AIDS onset- and rare CMV infection-associated cell subsets in peripheral blood, and extremely rare leukaemic blast populations in minimal residual disease-like situations with frequencies as low as 0.01%.

[1] Institute for Molecular Systems Biology, Department of Biology, ETH Zurich, Auguste-Piccard-Hof 1, Zurich 8093, Switzerland. [2] Swiss Institute of Bioinformatics, Zurich 8093, Switzerland. [3] Life Science Graduate School Zurich, PhD Program Systems Biology, Winterthurerstr. 190, Zurich 8057, Switzerland. Correspondence and requests for materials should be addressed to M.C. (email: claassen@imsb.biol.ethz.ch).

The health and disease status of multicellular organisms pivotally depends on rare cell populations, such as haematopoietic stem cells or tumour-initiating cell subsets[1]. Advances in single-cell-resolved molecular measurement technologies have increasingly enabled the description of cell population heterogeneity, including rare subpopulations, in health and disease[2]. It is becoming routine to measure thousands of DNA, RNA[3] and dozens of protein[4] species in thousands of single cells, optionally including their spatial context[5–7].

Such multiparametric single-cell snapshots have been used to define heterogeneous cell population structure using unsupervised clustering techniques that generate a *representation* of a cell population, defined in terms of cluster-based *features* such as cluster medians[8]. While *unsupervised machine learning* constitutes a powerful exploratory tool, the identification of disease-associated cell subsets requires a further *supervised learning* step to associate the clustering-derived representation with disease status. Unsupervised approaches have been extended to the classification of single-cell samples and have been successful where disease association manifested itself in condition-specific differences of abundant cell subpopulations[8,9].

Unsupervised approaches describe general population features that are not necessarily associated with disease status. Typically a large number of cell population features (thousands[9] or millions[10]) are required to detect rare cell subsets from high-dimensional measurements (i.e., 20+ dimensions). Most such features are not relevant, leading to *overfitting* or even precluding the identification of disease-associated rare cell populations. As this study will demonstrate, this situation severely limits the capacity of existing approaches to take advantage of novel highly multiparametric single-cell measurements to yield insights into the subpopulation-origin of diseases such as minimal residual disease (MRD) or tumour-initiating cells[1].

CellCnn overcomes this critical limitation and facilitates the detection of rare disease-associated cell subsets. Unlike previous methods, CellCnn does not separate the steps of extracting a cell population representation and associating it with disease status. Combining these two tasks requires an approach that (1) is capable of operating on the basis of a set of unordered single-cell measurements, (2) specifically learns representations of single-cell measurements that are associated with the considered phenotype and (3) takes advantage of the possibly large number of such observations. We bring together concepts from *multiple instance learning*[11] and *convolutional neural networks*[12] to meet these requirements.

In this study, we apply CellCnn in a classification setting to reconstruct cell-type-specific signalling responses in samples of peripheral blood mononuclear cells (PBMCs). We additionally apply CellCnn in a regression setting to identify abundant cell populations associated with disease onset after HIV infection, and achieve comparable prediction accuracy to a state-of-the-art analysis performed recently[9], however with computational cost reduced by several orders of magnitude. Finally, we demonstrate the unique ability of CellCnn to identify extremely rare phenotype-associated cell subsets by detecting memory-like natural killer (NK) cells associated with prior cytomegalovirus (CMV) infection and leukaemic blasts in MRD-like situations.

## Results

### CellCnn overview.
CellCnn takes as input a set of observations of cellular populations (multi-cell inputs) each associated with a phenotype, for example, patient blood or tissue samples with associated disease status or survival information. It is difficult to learn the molecular basis of this association since it possibly manifests itself by differences of *a priori* unknown cell subsets. To address this difficulty, CellCnn associates a multi-cell input with the considered phenotype by means of a convolutional neural network. The network automatically learns a concise cell population representation in terms of molecular profiles (*filters*) of individual cells whose presence or frequency is associated with a phenotype (Fig. 1a and see section Methods).

Convolutional neural networks were originally designed to process the two-dimensional structure of images and typically consist of one or more sets of convolutional and pooling layers[12,13]. We adapted the convolutional neural network architecture to process unordered multi-cell inputs. Image patches correspond to individual cell measurements. The output of the convolutional filter is evaluated for every cell measurement. The computation at the pooling layer consists of selecting either the maximum (max-pooling) or mean (mean-pooling) response within the multi-cell input[12,13]. Pooling is performed separately for each convolutional filter. Max-pooling computes the maximum response over all members of a multi-cell input for a particular filter, and thereby measures the presence of cells yielding high cell filter response. Mean-pooling evaluates the average cell-filter response of a multi-cell input, and thereby serves as an approximation of the frequency of the cell subset strongly responding to a specific filter. Finally, the pooling layer is connected to the output layer. For regression problems the output layer contains a single node, whereas for classification problems it contains one node per class. The output of the network predicts the sample-associated phenotype (e.g., disease condition, expected survival). Network training optimizes weights so that the network-predicted phenotypes match the true phenotypes.

Trained filter weights correspond to molecular profiles of relevant cell subsets and allow for assignment of the cell subset membership of individual cells (cell-filter response, Fig. 1b). In some cases, a cell subset selected by a filter may comprise more than one cell type, each being associated with the studied phenotype such as response to a stimulus. To detect such situations, density-based clustering with respect to all measured markers is performed on the group of cells selected by each filter. Finally, to identify the characteristic markers of each selected cell type, a quantitative score, based on the Kolmogorov–Smirnov test statistic, is derived for each marker, summarizing the difference in marker abundance distribution between the whole-cell population and the selected cell type.

### Detecting cell types responding to cytokine stimulation.
First, we applied CellCnn to a mass cytometry data set acquired from samples of PBMCs[14]. These samples were exposed to various paracrine agents and their proteomic responses recorded at the single-cell level with respect to 14 intracellular markers and 10 cell-surface markers characteristic of immune cell type. CellCnn was trained for each paracrine agent to classify stimulated and unstimulated samples using only the 14 intracellular markers. We investigated the cell type-specific filter responses and found very specific and sensitive enrichment of the cell types expected to specifically respond to the considered agent, i.e., differential response by monocytes and dendritic cells in the case of granulocyte–macrophage colony-stimulating factor (GM-CSF) exposure[15] (Fig. 1f and see Supplementary Fig. 1 for the remaining agents considered in ref. 14). For the GM-CSF stimulation experiment, CellCnn learned one filter positively associated with GM-CSF exposure (see Supplementary Fig. 2 for filter weights learned for this example as well as for the remaining agents) and this filter was used to compute the weighted sum of the abundance profile for each single-cell (cell-filter responses, Fig. 1c,f). It is possible that the cell subset

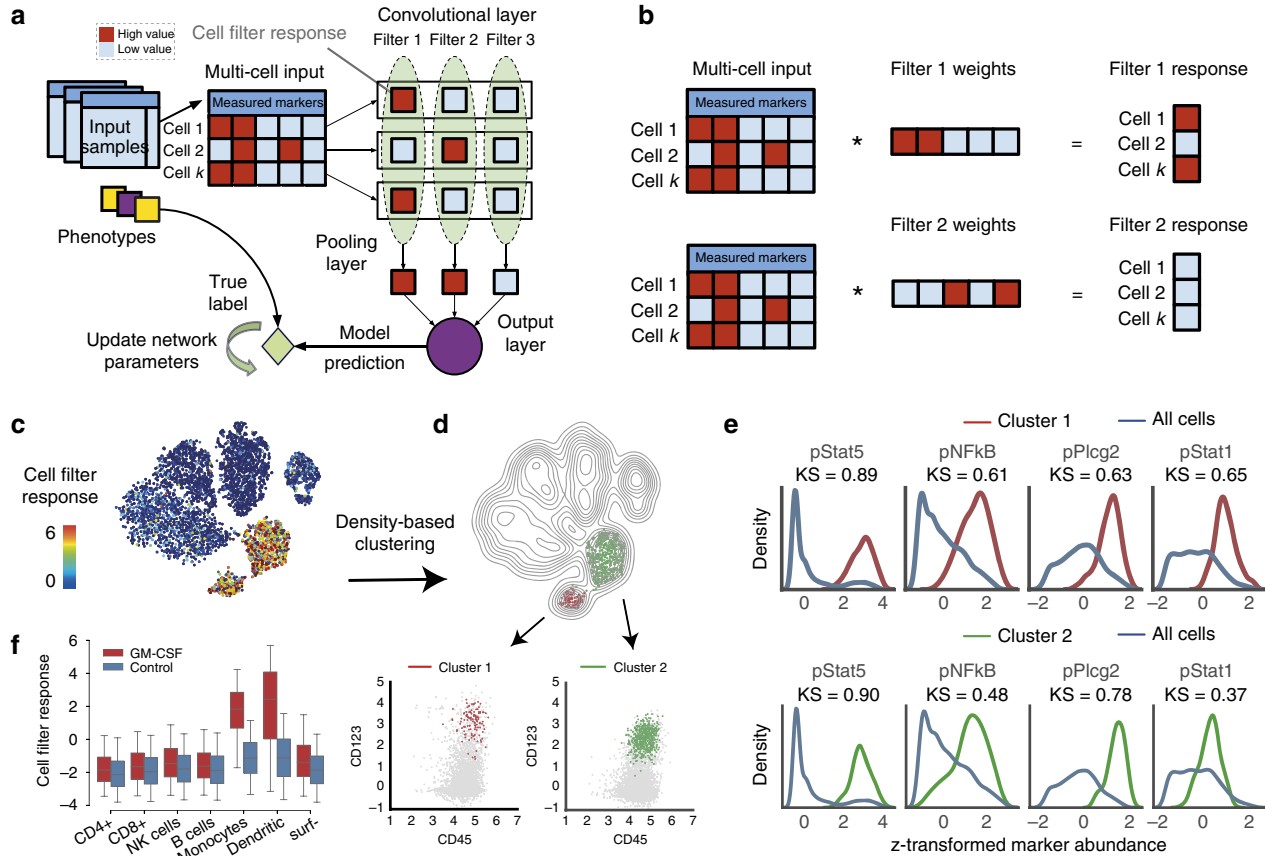

**Figure 1 | CellCnn overview and demonstration.** (**a**) CellCnn convolutional neural network architecture. CellCnn takes as input groups of single-cell measurements (multi-cell inputs), where each group is annotated with a phenotype. Node activities in the convolutional layer are defined as weighted sums over single-cell molecular profiles. Nodes in the pooling layer evaluate the presence (max pooling) or frequency (mean pooling) of specific cell subsets. The output of the network estimates the sample-associated phenotype (e.g., disease condition, expected survival). Network training optimizes weights to match training data set phenotype. Trained filter weights correspond to molecular profiles of relevant cell subsets and allow for assignment of the cell subset membership of individual cells (cell-filter response). (**b**) Illustration of cell-filter response computations for individual cells. For instance, marker profiles of cell 1 and 3 exhibit perfect/no match with weights of filter 1/2 and therefore result in a high/low (red/blue) cell-filter response. (**c**) CellCnn classification of GM-CSF (un-) stimulated peripheral blood mononuclear cell populations monitored with mass cytometry. t-SNE[28] projection based on all cell type-defining surface markers (not used by CellCnn), coloured by cell-filter response. (**d**) Density-based clustering of high cell filter-response regions using all cell-type-defining surface markers reveals two distinct cell types, namely monocytes (CD33+) and dendritic cells (CD123+). (**e**) Histograms of the signalling markers (used by CellCnn) showing greatest differential abundance in terms of the Kolmogorov–Smirnov two-sample test between the whole-cell population and the selected cell subsets. (**f**) Response of individual cells (grouped by manually gated cell types) is shown for both conditions. Significantly higher cell-filter response for monocytes and dendritic cells in the stimulated sample.

selected by this filter is characterized by a biological process that is active in various cell types. To identify such a situation we conducted additional density-based clustering of the group of cells exhibiting high cell filter response and found two distinct cell types, namely monocytes and dendritic cells (Fig. 1d). Both selected cell types exhibit, as to be expected, high pStat5 levels after GM-CSF stimulation (Fig. 1e).

**Detecting T-cell subsets prognostic of AIDS-free survival.** We used CellCnn to identify T-cell subsets associated with increased risk of AIDS onset in a cohort of 383 HIV-infected individuals[16]. Flow cytometry measurements of 10 T-cell-related molecular markers from peripheral blood and AIDS-free survival time were available for each individual. Trained on a subcohort of 256 individuals, CellCnn identified cell subsets with either elevated proliferation marker Ki67 or naive T-cell phenotype (Fig. 2b,c). The frequency of these cell subsets has been reported to be associated with AIDS-free survival in previous studies[9,10,17].

CellCnn was further used to categorize the remaining set of 127 test individuals into a low- and high-risk group (see Methods). Kaplan–Meier curves of these groups are significantly different (P-value = 3.03e-03, log-rank test; Fig. 2a). Citrus, a state-of-the-art approach to identifying clinically prognostic cell subsets[9] achieved a less significant dissection of the two risk groups on the same training and test data partition (P-value = 2.97e-02, Fig. 2a). CellCnn and Citrus identify the same strongly survival-associated cell populations (Supplementary Fig. 4). Furthermore, to assess the robustness of our approach, we reduced the size of the training cohort from 67 to 50% and 33% of the samples. In these more challenging settings, the stratification performance of CellCnn remained at equivalently high levels (Supplementary Fig. 3).

**Detecting rare NK-cell subsets associated with CMV infection.** We went on to assess CellCnn's capability to detect rare disease-associated cell populations. Specifically, we analysed

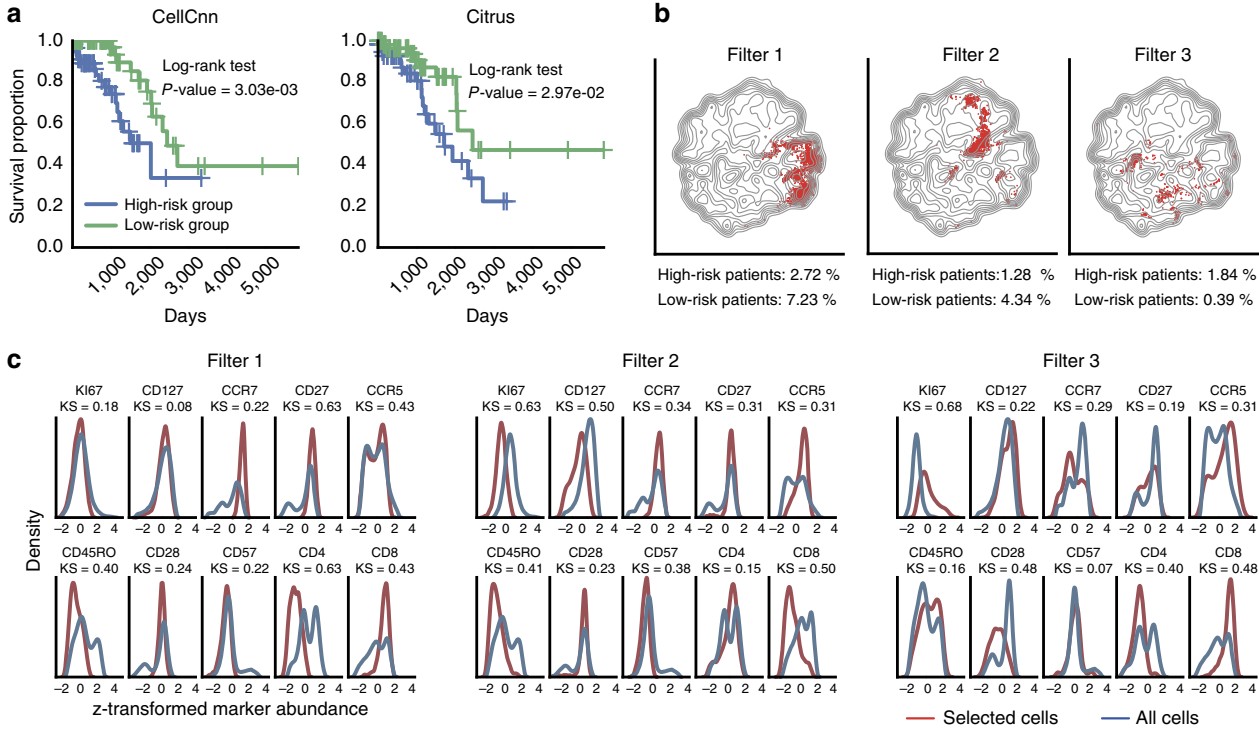

**Figure 2 | CellCnn analysis of immune cell populations associated with AIDS onset in HIV patients.** (**a**) Kaplan–Meier plots for high- and low-risk patient cohort according to CellCnn survival prediction ($P = 3.03e-03$, log-rank test, computation time: 1 h, single laptop core) and state of the art: Citrus ($P = 2.97e-02$, 3 days, 24 Intel Xeon cores). (**b**) Reconstruction of cell subsets predicting AIDS-free survival in HIV-infected patients. Cells selected by CellCnn filters are highlighted (in red) on the t-SNE map computed from all test samples. A distinct area is occupied by each selected subpopulation. Filters 1 and 2 are positively associated with survival, whereas filter 3 is negatively associated. Average frequency of the selected cell subsets in 10 test patients with lowest/highest survival times is reported. (**c**) Histograms of measured marker abundances for the whole-cell population and the selected cell subsets.

a mass cytometry data set acquired to characterize human NK cell diversity and associate NK cell subsets with genetic and environmental factors, namely prior CMV infection[18]. This data set comprises mass cytometry measurements of 36 markers, including 28 NK cell receptors, for PBMC samples of 20 donors with varying serology for CMV (see section Data sets in Methods). Applied to the ungated single-cell data, CellCnn identified two CMV seropositivity-associated cell populations (Fig. 3a and Supplementary Fig. 5). The most predictive cell population is rare (frequency < 1%), positively associated with previous CMV infection and exhibits a memory-like NKG2C+, CD57+ NK cell phenotype (Fig. 3b,c) as further described in ref. 18. The state-of-the-art cell population classifier Citrus failed to identify this rare, predictive cell population (Fig. 3a,b) and, as a result, exhibited inferior classification performance in comparison to CellCnn (Fig. 3d).

**Detecting rare blast cell populations in leukaemia.** We next assessed the scope of CellCnn to detect extremely rare cell populations associated with MRD in acute myeloid leukaemia (AML). Specifically, we analysed mass cytometry data sets of healthy bone marrow samples with *in silico* leukaemic blast spike-in subpopulations of decreasing frequency to mimic the MRD phenotype[19]. To objectively compare CellCnn with existing methods with respect to detecting rare phenotype-associated cell populations, we assembled a benchmark data set with clearly defined training/validation and test samples (see Data sets in Methods section). Spike-ins from patients characterized as cytogenetically normal (CN), as well as from patients with core-binding factor translocation [t(8;21) or inv(16)] (CBF) were

considered. CellCnn was trained on the three-class classification problem of sample stratification as healthy, CN AML or CBF AML and correctly identified the leukaemic blast subsets in the test samples (not used for training) at a frequency as low as 0.1% (500/500,000 blast/total cells) (Fig. 4a,b). We found that the predictive subsets for the AML subgroups shared differentially abundant markers (CD34, CD45, CD44) but also exhibited several differences (Fig. 4e). For instance, CN AML blasts were CD7+, CD38+, CD117+, whereas CBF AML blasts were CD15+, CD38mid. These results are in accordance with the findings presented in the original study[19].

Due to the limited number of test samples available, we assessed the ability of CellCnn to correctly predict the phenotype of new samples on the basis of the characteristics of the learned representation. A good representation should clearly separate healthy, CN AML and CBF AML samples. To this end, we computed a two-dimensional projection of each mass cytometry sample by projecting it to the two most relevant AML-specific filters. We refer to this projection as the CellCnn-based representation. In a similar fashion, we computed a two-dimensional Citrus-based representation by projecting each mass cytometry sample to the two most relevant AML-specific clusters. Finally, we derived two-dimensional moment-based and *autoencoder*-based sample representations by projecting the full sample representations to their first two principal components (for details see Methods). The two-dimensional representations for the training, validation and test samples obtained by the different methods are visualized in Fig. 5a,b, where it is illustrated that the CellCnn-based representation achieves the clearest separation between the healthy, CN AML and CBF AML samples.

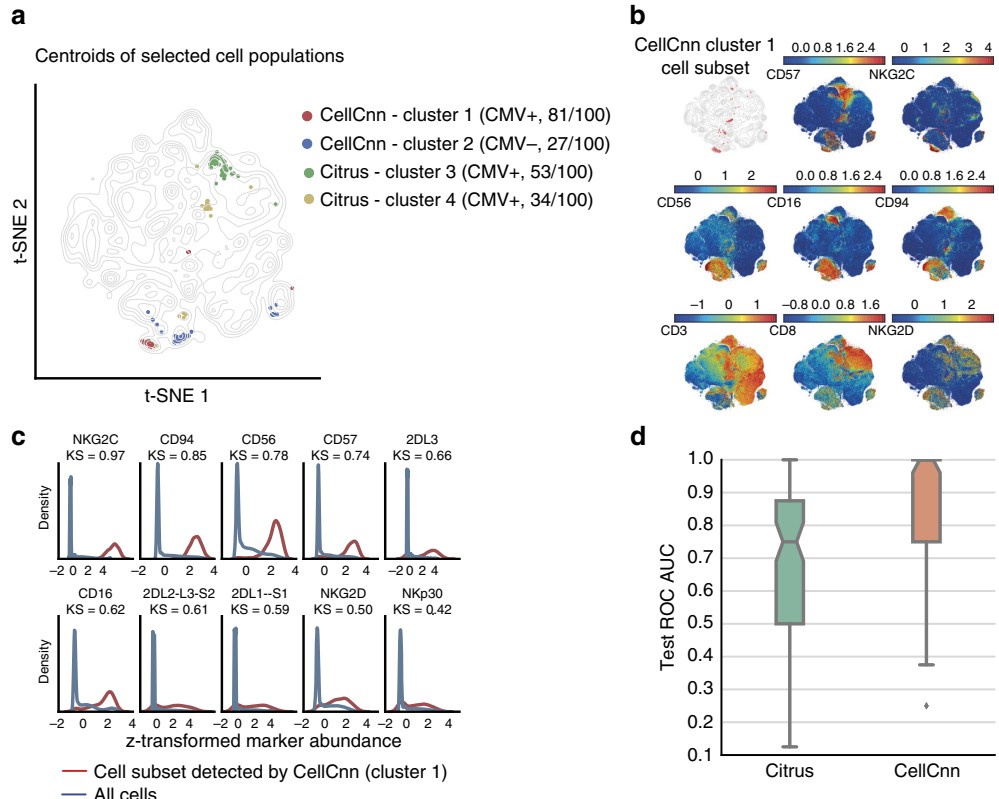

**Figure 3 | Detection of rare CMV seropositivity-associated cell populations. (a)** Visualization of the cell subsets selected by CellCnn and Citrus across 100 Monte Carlo cross-validation (CV) repetitions. Centroids of selected populations are highlighted on a t-SNE map computed from all samples using 20,000 cells per individual (see Methods for details). The cell population most frequently (81 out of 100 times) selected by CellCnn is positively associated with CMV prior infection, whereas the second most frequent cell subset is negatively associated with CMV seropositivity. **(b)** t-SNE map colour-coded according to abundance of selected markers. The top-left subplot depicts the cell subset most frequently selected by CellCnn, corresponding to cluster 1 in **a**, (see Methods for details). This cell subset corresponds to a memory-like (NKG2C+, CD57+) NK (CD56+, CD3−) and NK T (CD56+, CD3+) cell population. **(c)** Histograms of selected marker abundances for the whole-cell population and the cell subset most frequently selected by CellCnn. **(d)** Boxplot of area under the ROC curve (ROC AUC) on the test samples for 100 Monte Carlo CV repetitions. The median test ROC AUC for CellCnn is equal to 1.

Additionally, CellCnn was used for single-cell classification, i.e., to identify individual cells constituting the disease-associated cell subset. We compared CellCnn with (1) a state-of-the-art distance-based outlier detection algorithm[20], constituting a quantifiable variant of visually inspecting condition-specific projection map differences (e.g., t-SNE maps[21,22]); (2) logistic regression, support vector machine and random forest classifiers that take as input single-cell profiles; (3) Citrus[9]; and (4) single marker cutoff gates, all showing inferior performance at identifying the leukaemic blast subsets (Fig. 5e, Supplementary Fig. 7, for details see Methods). We further considered more extreme situations with decreasing frequency of the blast spike-in cell subset down to 0.01% (50/500,000 blast/total cells) (Figs 4c,d and 5a,d). While the task of recovering the correct cell subset becomes increasingly difficult, both the whole-sample representation learned by CellCnn as well as its single-cell classification precision stayed largely unchanged for all considered blast spike-in subset frequencies.

The above study is designed to identify disease-associated cell subsets that generalize to new patients not seen during CellCnn training. We further evaluated a personalized-medicine scenario where the same patient's bone marrow samples are assessed with respect to coherent cell population changes across different conditions. On the basis of the AML and acute lymphoblastic leukaemia (ALL) samples provided in refs 19,21 we constructed sample pairs of healthy bone marrow and

AML/ALL blasts of decreasing frequency. For this setting, CellCnn is able to faithfully recover leukaemic blast populations down to frequencies of 0.005% (Supplementary Methods and Supplementary Fig. 8).

## Discussion

CellCnn achieves this unprecedented high precision in the rare cell type setting by overcoming the inherent limitations of the unsupervised feature engineering strategies of state-of-the-art approaches. When analysing samples from modern single-cell techniques with increasing multiparametricity, such as mass cytometry, these approaches enumerate thousands[9] or an exponential number[10] of features, at the cost of accumulating many potentially uninformative, confounding features. This situation leads to both computational bottlenecks and loss of statistical power. CellCnn provides a solution to this limitation by jointly and thereby efficiently solving the feature engineering, selection and association tasks in a single supervised learning step.

CellCnn analysis allows for description of the molecular makeup of phenotype-associated cell populations. While a direct interpretation of the learned filter weights might not be sufficient for defining properties of the phenotype-associated cell populations, we suggest using the learned filters to select and interpret the phenotype-associated cell populations in the

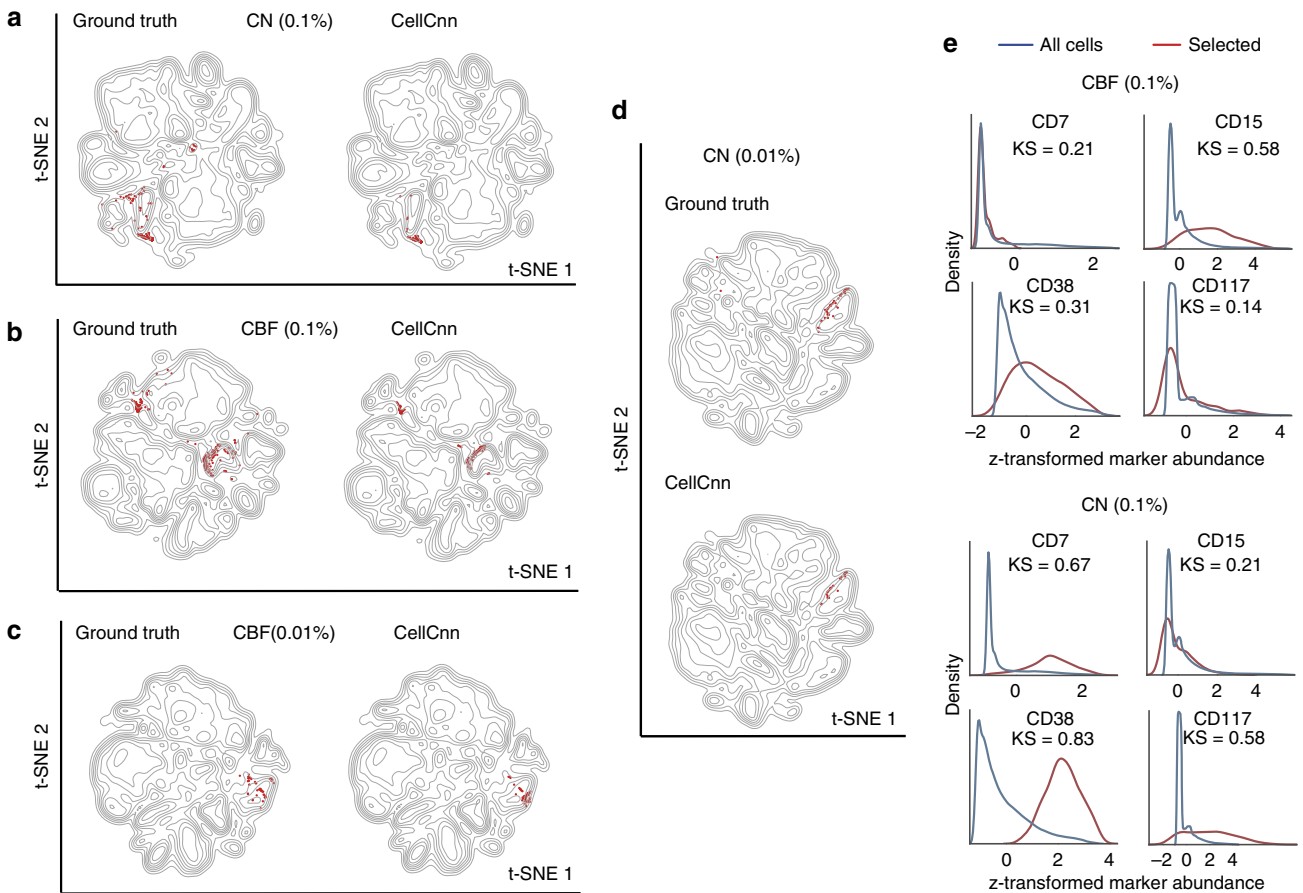

**Figure 4 | Identification of *in silico* spike-in rare leukaemic blast populations for two AML subgroups.** (**a**) The spiked-in subset (frequency = 0.1%) of blast cells from a cytogenetically normal (CN) patient is highlighted in red on the left plot (ground truth) and compared with cells identified by CellCnn, which are marked in red on the right plot. (**b**) Similar setting as (**a**) for a spiked-in subset of blast cells from a core-binding-factor translocation (CBF) patient. (**c,d**) Similar settings as (**a,b**) for spiked-in subsets of blast cells with even lower frequency (0.01%). (**e**) Histograms of selected cell surface markers for the disease-associated cell populations identified by CellCnn. The markers presented highlight the differences of blast cell immunophenotypic profiles between CBF and CN patients. CBF, core binding factor translocation; CN, cytogenetically normal.

analysed single-cell data. Specifically, we evaluate the underlying potential multiplicity of cell types by clustering and report statistics such as histograms on their marker profiles. It is conceivable that CellCnn does not identify the entirety of phenotype-associated cell types. Such a situation can arise if the association with the phenotype manifests itself by the activity of multiple processes spanning various distinct cell subsets, as it is the case for AIDS onset in HIV patients. In this case, CellCnn's regularization mechanisms will favour a simple and yet effective association via a fraction of these cell subsets. Iterative application of CellCnn with explicit exclusion of already selected cell populations could be explored as a means to identify further redundant, but possibly biologically relevant, phenotype-associated cell populations. However, iteratively applying CellCnn to the HIV cohort data set did not reveal further AIDS onset-associated cell populations.

CellCnn can be used in conjunction with other cell population detection methods such as Citrus. Since these follow a conceptually different approach, detection of a cell population by both methods would be a strong indicator of this population's association with the studied phenotype. However, in addition to rare cell types, further situations are conceivable where CellCnn would correctly detect a cell population that Citrus does not detect. In particular, CellCnn should be better at finding cells that do not form distinct subset clusters in

the space defined by all measured markers (Supplementary Fig. 9).

Furthermore, neural network training is efficient, scaling linearly with the number of measured components. Consequently, CellCnn is applicable to a variety of highly multi-parametric single-cell data sources beyond flow and mass cytometry, such as single-cell RNA sequencing or imaging data. In this study, we have demonstrated the ability of CellCnn to *ab initio* identify phenotype-associated cell subsets in publicly available data sets with known ground truth. Given the expected increase in patient cohort sizes in concerted initiatives as the The Cancer Genome Atlas (TCGA) and concomitant rise in their analysis with single-cell technologies[23], we expect scalable representation learning approaches such as CellCnn to uniquely take advantage of the resulting data by enabling the discovery of disease mechanisms mediated by rare cell populations, in both basic research and personalized medicine.

## Methods

**Data sets.** The mass cytometry data set of PBMCs and the flow cytometry data set of HIV-infected patients were adopted, respectively, from Bodenmiller *et al.*[14] and the U.S. Military HIV Natural History Study[16].

The mass cytometry data set for the first rare cell type study is based on the data set from Horowitz *et al.*[18]. All analyses were performed on the ungated PBMC

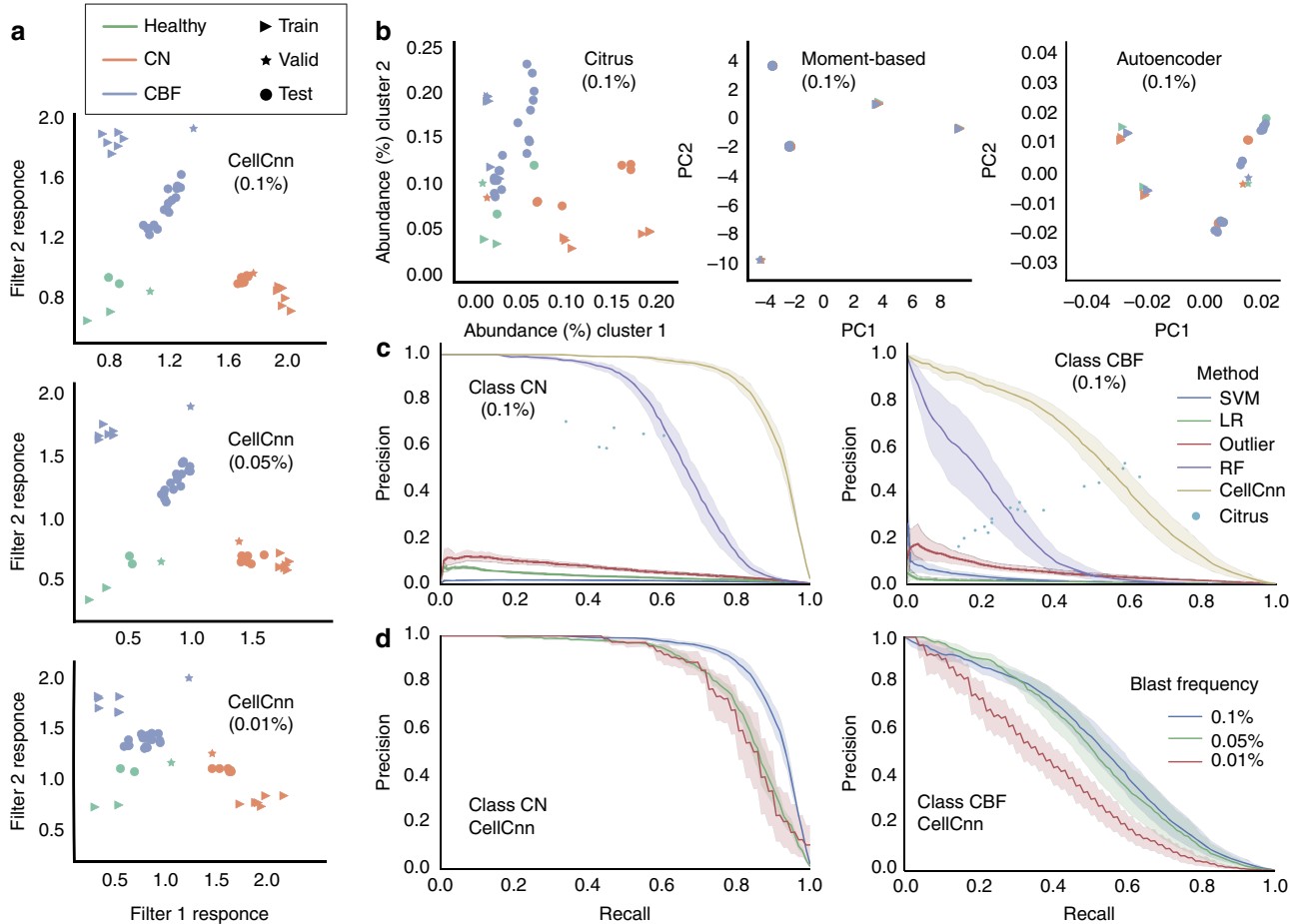

**Figure 5 | Benchmark results on the identification of *in silico* spike-in rare leukaemic blast populations for two AML subclasses. (a)** Whole-sample representation learned by CellCnn for various AML blast cell population frequencies. The three classes are well separated (linearly separable) in the CellCnn-based representation space (i.e., when projected to the two most relevant AML-specific filters). **(b)** Comparison to the baseline methods for whole-sample representation (Citrus[9], moment-based: multi-cell input summary profiles, denoising autoencoder[26]) for AML blast population at 0.1%. The three classes are not well separated in the representation space learned by these approaches. **(c)** Comparison to baseline methods for single-cell classification (LR, logistic regression; outlier, distance-based outlier detection[20]; RF, random forests; SVM, support vector machines Citrus[9]) for AML blast population at 0.1%. For all methods except Citrus, average precision–recall curves for recovery of blast cells on the test samples are reported. Shadowed areas indicate 95% confidence intervals. Citrus does not provide a precision–recall series; therefore, a single precision–recall point is computed for each test sample. **(d)** Single-cell classification performance of CellCnn for various low AML blast cell population frequencies. Average precision–recall curves on the test samples are reported with shadowed areas indicating 95% confidence intervals. CBF, core binding factor translocation; CN, cytogenetically normal.

samples after filtering out dead cells and doublets as described in ref. 18. The data set comprises PBMC samples from 21 individuals with associated CMV serology status. Sample 008 was described in the original study[18] as an outlier with respect to memory-like NK cell population abundance and was therefore excluded from our benchmark analysis. The remaining samples (11 CMV −, 9 CMV +) were randomly split into training and test sets using Monte Carlo cross validation (CV). The mass cytometry data set for the second rare cell type study is based on the healthy and AML bone marrow samples provided by Levine *et al.*[19] We focused on the AML samples with at least 10% CD34 + blast cells (gating depicted in Supplementary Fig. 12) and for which additional cytogenetic information was available. More specifically, patients SJ10, SJ12, SJ13 were characterized as CN, whereas patients SJ1, SJ2, SJ3, SJ4, SJ5 presented core-binding factor translocations [t(8;21) or inv(16)] (CBF). On the basis of these samples, we assembled a benchmark data set for the three-class classification problem of sample stratification as healthy, CN AML or CBF AML. Below we describe the benchmark data set generation strategy.

The *training set* comprises cells from two healthy bone marrows (BM1, BM2) as well as from patients SJ10 (CN) and SJ1 (CBF). Cells from each healthy BM were split at random into two mutually exclusive sets: (BM1a, BM1b) and (BM2a, BM2b), respectively. BM1a and BM2a are the training samples from the healthy class. BM1b and BM2b are used to create synthetic MRD samples. For each patient, we randomly drew six subsets of blast cells and computationally mixed each of these subsets with either BM1b or BM2b. In total, we have two healthy samples, six samples from class CN and six samples from class CBF.

The *validation set* comprises cells from one healthy bone marrow (BM3) as well as from patients SJ12 (CN) and SJ2 (CBF). Cells from BM3 were split at random into two mutually exclusive sets: (BM3a, BM3b). BM3a is the validation sample from the healthy class. BM3b is used to create synthetic MRD samples. For each patient, we randomly drew one subset of blast cells and mixed it with BM3b. In total, we have one healthy sample, one sample from class CN and one sample from class CBF.

The *test set* comprises cells from two healthy bone marrows (BM4, BM5) as well as from patients SJ13 (CN), SJ3 (CBF), SJ4 (CBF) and SJ5 (CBF). BM4 and BM5 are the test samples from the healthy class. BM4 and BM5 are also used to create synthetic MRD samples for testing. For each patient, we randomly drew six subsets of blast cells and mixed each of these subsets with either BM4 or BM5. In total, we have 2 healthy samples, 6 samples from class CN and 18 samples from class CBF. Using the procedure described above, three benchmark data sets were created with blast spike-in frequencies of 0.1% (500 cells), 0.05% (250 cells) and 0.01% (50 cells).

The mass cytometry data set for the 'personalized medicine' study is based on the healthy and AML bone marrow samples provided by Levine *et al.*[19] and on the ALL samples provided by Amir *et al.*[21]. MRD-like samples were created by computationally combining CD34 + gated AML/ALL blast cells and cells from a healthy bone marrow sample into mixed synthetic samples. Different mixed samples were created by combining different numbers of blast cells from different patients with cells from each healthy bone marrow sample (five matching healthy bone marrow samples were available for AML and one healthy bone marrow sample for ALL). The considered numbers of blast cells are 1,000, 500, 250, 100, 50

and 25. In each case, cells from the healthy bone marrow were split at random into two mutually exclusive subsets. The first subset was used to create the MRD-like sample, while the second one served as control sample for training CellCnn.

**CellCnn network architecture and training.** CellCnn takes as input groups of single-cell measurements (e.g., mass cytometry samples), each group annotated with a phenotype and aims at identification of phenotype-associated cell subpopulations. This is an example of a *multiple instance learning* task[11] and it is addressed with a convolutional neural network approach. CellCnn implements a variant of a convolutional neural network. Such networks are artificial neural networks originally designed to process the two-dimensional structure of images and typically consist of one or more sets of convolutional and pooling layers[12,13]. Briefly, the convolutional layer comprises filters that evaluate the occurrence of specific patterns in image patches and the pooling layer computes summaries of these occurrences. We adapted the convolutional neural network architecture to process unordered multi-cell inputs. Image patches correspond to individual cell measurements. Each cell measurement was evaluated with respect to every convolutional filter, i.e., to its fit to respective molecular profile (cell-filter response) in the convolutional layer. The computation at the pooling layer consisted of selecting either the maximum (max-pooling) or mean (mean-pooling) response within the multi-cell input. Pooling was performed separately for each convolutional filter. Max-pooling computes the maximum response over all members of a multi-cell input for a particular filter, and thereby measures the presence of cells yielding high cell filter response. Max-pooling was performed for the analysis of the peripheral blood and AML data sets, where cell presence appeared to be most informative. Mean-pooling evaluates the average cell-filter response of a multi-cell input, and thereby serves as an approximation of the frequency of the cell subset strongly responding to a specific filter. Cell subset frequencies turned out to be most informative for the analysis of the HIV data set and therefore mean-pooling was performed. Finally, the pooling layer was connected to the output layer. For regression problems the output layer contains a single node, whereas for classification problems it contains one node per class. Nodes in the output layer compute a weighted sum over the pooling layer nodes, followed by a nonlinear operation (hyperbolic tangent for regression and softmax for classification, further details in Supplementary Methods).

The convolutional filter weights and output layer weights were optimized for optimal association of multi-cell inputs with their phenotype labels using mini-batch stochastic gradient descent with Adam[24] updates. Random search was used to optimize a set of important hyperparameters (number of filters, learning rate, dropout), whereas the remaining ones were kept fixed during all experiments (mini-batch size = 128, maximum number of training epochs = 100, early stopping after 5 epochs, $L_2$ weight decay = 1e-04). Network weights were randomly initialized from a uniform distribution. Mean squared error was minimized for regression problems and categorical cross entropy for classification (see for more details on the CellCnn methodology in Supplementary Methods.)

**Characterization of identified cell subsets.** Trained network filters are used to select phenotype-associated cell populations in the analysed single-cell data. In our experiments, we used a cutoff threshold of 0.5 × (maximum cell-filter response achieved for a particular filter) in order to define filter-specific cell subsets. It is expected that, in some cases, a filter-specific cell subset may comprise more than one cell type. To resolve such a situation, density-based clustering using the DBSCAN algorithm[25] is performed on the group of cells selected by each filter. Finally, for each cluster identified by DBSCAN, we quantified the differences between univariate marker distributions of the whole-cell population and the cluster-specific cell population via the Kolmogorov–Smirnov two-sample test statistic.

**Cell subsets as multi-cell inputs.** To take advantage of high-content single-cell techniques like flow or mass cytometry, CellCnn optionally takes multiple random cell subsets of a specific cytometry sample as input to increase the effective number of data points for association.

In all our experiments, random cell subsets, drawn with replacement from the original cytometry samples, were used as multi-cell input training examples of CellCnn. In all cases, we generated an equal number of multi-cell inputs associated with each label. The number of multi-cell inputs and the number of cells in each multi-cell input were chosen on the basis of the validation set. Our experiments on mass cytometry data showed that CellCnn is not very sensitive to these two hyperparameters, as long as the multi-cell inputs are chosen sufficiently large to contain cells with the molecular profile of interest (Supplementary Fig. 10). To detect extremely rare populations (abundance < 0.1%), we used a modified procedure for creating multi-cell inputs. Fifty per cent of a multi-cell input was sampled uniformly at random from the whole-cell population whereas the other 50% was sampled from cells with high *outlierness* score. We define the outlierness score of each cell on the basis of the distances between this cell and its closest neighbours from the control samples[20] (details in Supplementary Methods).

**Model selection and interpretation.** *PBMC data set.* Each sample was initially split into a training (80%) and a validation (20%) set of cells. We trained 10 models,

each comprising two filters, with initial weights drawn from a uniform distribution and selected the model with highest predictive accuracy on the validation set. The filter with highest weight connection to the output node corresponding to stimulation was chosen for the detection of stimulated cells.

*HIV-cohort data set.* The full patient cohort was randomly split into a training (2/3) and a test (1/3) cohort. We used fivefold CV on the training cohort resulting in five models, each trained on a different subset of the training cohort. For each CV fold, random search was used to optimize over different hyperparameter settings and the model achieving best predictive performance on the corresponding validation samples was chosen. The hyperparameters finally adopted are the following: 3–5 filters (varying among CV folds), no dropout regularization, learning rate = 0.01. Finally, an ensemble model, consisting of the five best networks (one from each CV fold), was used to predict survival times for the individuals in the test cohort. For the test phase, one subset of 3,000 cells was used per individual. The output of CellCnn corresponded to predicted disease-free survival time for each patient and was used to split the test cohort into a low-risk and a high-risk group. The threshold used for defining the two groups was the median predicted survival time. The survival distributions of the low- and high-risk groups were compared using a log-rank test, as the data set contained several right-censored observations.

Information from all CV runs was used to select frequently occurring filters. We compiled a matrix of all filter weights from the five networks and performed hierarchical clustering using cosine similarity as metric (Supplementary Fig. 3c). For each cluster of filters with at least two members, a cluster-specific cell population was defined as the intersection of the sets of cells selected by the filters belonging to that cluster. The reported phenotype-associated cell populations correspond to the cluster-specific cell populations defined using the above strategy. To assess the robustness of our approach, the same procedure was repeated using 50 and 33% of the samples for training and, correspondingly, 50 and 67% of the samples for testing (Supplementary Fig. 3b,d).

*NK-cell benchmark data set.* A Monte Carlo CV procedure was used to randomly split the available mass cytometry samples (11 CMV −, 9 CMV +) into training and test sets. We performed 100 Monte Carlo CV repetitions, each time using 7 CMV − and 7 CMV + samples in the training set. The predictive performance of each trained model was evaluated on the basis of the six left-out samples (test set). The area under the ROC curve for the test set (test ROC AUC) was used as metric to quantify classifier performance. The test ROC AUC values from the 100 Monte Carlo CV repetitions are summarized in the boxplots of Fig. 3d.

Within each Monte Carlo CV repetition, the corresponding training set was further split into a training and a validation set using a threefold CV scheme (we refer to this scheme as nested CV). For each nested CV fold, random search was used to optimize over different hyperparameter settings and the model achieving best predictive performance on the corresponding validation samples was chosen. The hyperparameters finally adopted are the following: 3,000 cells per multi-cell input, 200 multi-cell inputs per sample, 3–5 filters (varying among CV folds), no dropout regularization, learning rate = 0.01, mean-pooling of the top 1% of cells in each multi-cell input. The network achieving the highest validation accuracy in the nested CV runs was used for the final predictions on the test set. For the test phase, one subset of 20,000 cells was used per individual.

To interpret the trained models, we sought to identify cell populations frequently selected by the CellCnn filters across the 100 Monte Carlo CV runs. For each filter, a filter-specific cell subset was defined on the basis of cell-filter responses, using a cutoff threshold of 0.5*(maximum cell-filter response achieved for this filter). Each filter-specific cell subset is compactly represented by its centroid. Therefore, we computed filter-specific centroids and used hierarchical clustering to group these centroids into clusters. By clustering the filter-specific centroids, we effectively assigned the filter-specific cell populations into similarity groups. For each cluster of centroids, we then computed the number of occurrences of any of its members in the 100 Monte Carlo CV runs. We followed a similar procedure to analyse the trained Citrus models: we computed the centroids of cell clusters with non-zero logistic regression coefficients, grouped these centroids via hierarchical clustering and counted the number of occurrences of each cluster in the 100 Monte Carlo CV runs. For both CellCnn and Citrus, we reported the centroids belonging to clusters that occurred in at least 20 of the 100 Monte Carlo CV repetitions. Centroids were projected to a t-SNE map obtained from all samples by computing the nearest neighbour cell in marker space and using the projection of that cell as the projection of the centroid (Fig. 3a). For CellCnn, the most frequently occurring cluster was selected in 81 out of 100 Monte Carlo CV runs. To further characterize this frequently selected cluster, we chose a representative centroid (centroid with minimum sum of distances to the other cluster members) and extensively characterized the corresponding cell population from which the centroid was computed (Fig. 3b,c). The same analysis was repeated for the second most frequently occurring cluster (Supplementary Fig. 5).

*AML benchmark data set.* The training, validation and test sets were defined as described previously (see section Data sets in Methods). Training samples were used to fit the model, validation samples were used to optimize over hyperparameters using random search and test samples were used to evaluate the generalization performance of the best fitted model. The hyperparameters finally adopted are the following: 20 filters with dropout regularization, learning rate = 0.01.

The validation samples were used to select AML-specific filters from the best performing model. For each AML subgroup (CN and CBF), we selected the filter achieving the highest difference of maximum cell-filter responses (averaged over the top 30 cells) between healthy and AML subgroup validation samples. Subsequently, the selected filters were used to (a) obtain a representation of each sample in terms of maximum cell-filter response, averaged over the top 30 cells (Fig. 5a,b) compute precision–recall curves for the test samples (Fig. 5c,d).

**Baseline methods.** The following models were used for comparison with CellCnn.

*Outlier detection.* We used a state-of-the-art distance-based outlier detection method[20]. A set $S$ of $s$ observations (single-cell profiles) is randomly sampled from the *inlier* class (i.e., the healthy control samples) and then used to evaluate the *outlierness* of single-cell profiles in the test samples. The outlierness of an observation is defined as the $L_1$ distance between this observation and its closest neighbour in $S$. Results for different values of $s$ are given in Supplementary Fig. 11. We finally used $s = 200,000$.

*Single-cell input logistic regression/SVM/random forest.* A logistic regression/support vector machine/random forests classifier that takes as input single-cell profiles from the multi-cell inputs generated for CellCnn. Each single-cell profile is labelled with the label (e.g., disease condition, survival time) of its corresponding cytometry sample. Random search was used to optimize over the hyperparameters of the classifiers.

*Moment-based representation.* A moment-based summary of a set of single-cell measurements. The first four moments of the marker abundance distributions are computed.

*Denoising autoencoder.* An unsupervised representation learning model[26] that is trained to reconstruct the original input from a corrupted version of it, e.g., after addition of Gaussian noise. In our experiments, we used the same multi-cell inputs and network architecture as for CellCnn, but removed the pooling layer and substituted the output layer by the multi-cell input. Random search was used to optimize over the number of filters and the standard deviation of Gaussian noise added to the input.

*Citrus.* A state-of-the-art approach for detecting phenotype-associated cell subpopulations[9]. Citrus initially performs hierarchical clustering of single-cell profiles from all considered cytometry samples, selects the clusters that contain at least a minimum number of cell events (according to the minimum cluster size threshold that is defined by the user) and computes cluster-based features (e.g., population medians or abundances) individually for each cytometry sample. The computed cluster-based features are used as input to an $L_1$ regularized predictor that detects phenotype-associated differentially abundant features. See Supplementary Methods for a detailed description of the parameters used in individual Citrus runs.

**Computation of two-dimensional sample representations.** Due to the limited number of test samples available, the ability of different methods to correctly predict the phenotype of new samples was assessed on the basis of the characteristics of the learned whole-sample representation. A good representation should clearly separate healthy, CN AML and CBF AML samples. For CellCnn and the denoising autoencoder, a sample representation was computed as the vector of maximum cell-filter responses, averaged over the top 30 cells. For Citrus, a sample representation was computed as the vector of cluster abundances for all clusters with non-zero coefficients, as computed by the $L_1$ regularized classifier. For the moment-based sample representation, we used the first four moments of the marker abundance distributions.

The high-dimensional representations computed by the different methods were visualized via two-dimensional projections on the first two principal components (see Fig. 5b for the moment- and autoencoder-based PCA projections, Supplementary Fig. 6 for the CellCnn- and Citrus-based PCA projections). Additionally, for CellCnn and Citrus we computed a more intuitive two-dimensional projection. Instead of projecting a feature vector to the first two principal components, we projected it to the two most relevant features (maximum cell-filter responses for CellCnn, cluster abundances for Citrus) (Fig. 5a,b). The two most relevant AML-specific filters/ clusters are selected on the basis of the validation samples. For each AML subgroup (CN and CBF), the filter achieving the highest difference of maximum cell-filter responses (averaged over the top 30 cells) between healthy and AML subgroup validation samples was selected.

**Machine learning glossary.** Technical terms are displayed in italic font when first introduced and are exemplified in Supplementary Note 1.

**Code availability.** CellCnn is implemented in Python 2.7 and uses the neural network libraries Theano[27] and Keras (https://github.com/fchollet/keras).

It is available for download at http://www.imsb.ethz.ch/research/claassen/ Software/cellcnn.html.

**Data availability.** This study uses previously published data sets. These data sets are available at http://www.imsb.ethz.ch/research/claassen/Software/cellcnn.html. The corresponding studies are cited in the respective sections.

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

## Acknowledgements

E.A. is funded by the SystemsX.ch RTD project PhosphonetPPM. We thank Justin Feigelman, Will Macnair and Uwe Sauer for helpful comments on the manuscript.

## Author contributions

M.C. and E.A. conceived the CellCnn methodology, and designed experiments. E.A. implemented the CellCnn method and carried out the experiments. M.C. and E.A. wrote the manuscript.

**Additional information**

**Competing interests:** The authors declare no competing financial interests.

**Publisher's note**: 

