## [Peer review file · Nature Communications]

Reviewers' comments:

Reviewer #1 (Remarks to the Author):

In this manuscript, Arvaniti and Claassen present a tool called CellCnn designed for representation learning. A key element of this supervised approach is that CellCnn learns patterns for cells specifically associated with labels of prior knowledge. The algorithm can then test how a given cell in a test sample "responds to the filter" (i.e. does it closely match the filter or not). Applications explored included: 1) learning a classifier for unstimulated and GM-CSF stimulated peripheral blood mononuclear cells, 2) learning a classifier for cells associated with risk of AIDS onset in HIV infected individuals and using this information to categorize patients in a way that stratified survival, and 3) learning a classifier for leukemia cells spiked into healthy bone marrow. New biological findings are not presented. The authors conclude that this tool "will enable discovery of new disease-associated populations."

Notes:

While the topic of computational analysis of single cell data is likely of significant interest to a broad range of researchers, it may be difficult for biologists to interpret the findings of CellCnn. In particular, the filters do not necessarily correspond directly to cell populations and it would be useful to explore this more as this is both a strength of the algorithm and a potential pitfall for reading and interpreting the findings. If the tool can use cellular information to stratify risk or diagnose disease, then it may not matter if the filters match "true populations." However biologist readers could use guidance on how to interpret the findings (what further analysis is needed to know if cells are one or more population?), and it may be confusing for some to try to interpret the filters and response patterns. Critically, cells responding strongly to filters may contain mixtures of distinct populations. For example, in Figure 1b, the cells responding to the classifier belong to at least 2 different populations (monocytes and dendritic cells). Significant additional analysis and discussion of this would strengthen the manuscript.

In Figure 1e, Filter 2 has both CD4 and CD8 as red for the filter (high) and Filter 1 has both CD4 and CD8 as blue (low). This is confusing. Are cells responding to Filter 2 double positive T cells (CD4+ CD8+ Ki67+)? Or are two different populations of cells able to respond to Filter 2 (CD4+ CD8- Ki67+ cells and CD4- CD8+ Ki67+ cells)? Or is this a typographical error? The histograms do not appear to be consistent with the heatmap, as CD4 appears to be higher in Filter 1 matching cells (red line vs. blue) and lower in Filter 2 matching cells (green line vs. blue), whereas CD8 has the opposite pattern. The histograms make more sense that the heatmap below, where CD4 and CD8 are both red in Filter 2.

How is it assessed whether the filters are over-fit to the training data? Would the filter in Figure 1c function effectively with only p-STAT5? How many other features in the filter are necessary?

It would be helpful to determine whether the computational approach is better than drawing simple cutoff gates, especially for the leukemia data in Figure 2.

80% of the data was used to train the filters and 20% to test. How does it perform working with a smaller amount of training data (e.g. 33%) and the remaining data in the test setting?

How different are the biological features used by CellCnn and Citrus to stratify in Figure 1f? Are they finding essentially the same cells, or very different cells? Would ensemble analysis be beneficial? Is either tool more likely to find a particular type of cell? Perhaps CellCnn is better at finding cells that do not form distinct subset clusters?

For AML, a concern with supervised approaches is over-fitting to the phenotype of the AML cells seen at diagnosis. The phenotype of the leukemia blasts changes following treatment and relapse, and there may be emergence of non-leukemic stem/progenitor cells that could confound the analysis. The "spike in" is not a sufficiently realistic test, although it is a good first step. An important comparison would be to show that CellCnn can be trained on pre-treatment samples of leukemia and then still function well in real post-treatment samples to distinguish leukemia blasts / MRD from recovering healthy stem/progenitor cells.

Presentation of an example where this approach reveals new biological findings would significantly strengthen the manuscript. It is not clear whether this approach simply provides a faster way to achieve things that are currently achieved with other tools. Does the emphasis of single cell vs. population learning provide a fundamentally new and better way to achieve cell classification?

Organization and other points:

Figure 1 has three distinct parts: 1) the outline of the method, 2) the GM-CSF signaling data, and 3) the HIV data and comparison to Citrus. It may make sense to split these up, especially GM-CSF and HIV.

In Figure 1e, the features in the heatmap appear to be scaled to different scales for each feature (e.g. Ki67 is as pink as CD8, despite being only weakly enriched in cells with a response to Filter 2). It may be different in Figure 1c (where pStat5 appears to set a uniform scale). The authors should make it clear how the color values / scales in the heatmap are set up and keep them consistent throughout when possible.

The manuscript may be challenging for biologist readers to follow in the current form as it uses field specific language. It would be stronger if a wider audience could read it, as machine learning tools for single cell data will be of general interest.

Reviewer #2 (Remarks to the Author):

The paper describes the application of convolutional neural networks for learning predictive feature representations for (rare) cell sub types that are associated with disease using single-cell markers. There has been a recent interest in using neural network architectures and deep learning for addressing important problems in genomics and computational biology. The proposed method follows this trend and considers a biologically interesting and relevant application of deep learning to single-cell biology.

While this work is conceptually interesting, I am not convinced that the results are sufficiently strong to warrant publication in a high-profile journal. The prediction performance, where rigorously evaluated using training/validation and test sets, appears to be comparable to existing methods. Major reported advantages are obtained in the regime of extremely rare cell types. I feel that the validation approach used for this task in particular is not sufficiently transparent and lacks rigor. Instead of using independent test sets to quantify model performance, the authors consider a validation set during training, in some instances directly to select filters, prior to reporting prediction accuracies on the same dataset. This approach, while I understand its motivation, makes it impossible to objectively compare and quantify the performance of alternative methods. The risk of overfitting when using deep models is large, simply because these models are highly parameterized.

More work will be needed to convincingly show that the results obtained on rare cell types are real and practically relevant. This would most likely require additional datasets and further experiments.

Minor comments:

Deep learning methods are expected to result in the practical performance gains if representations from high-dimensional feature sets are extracted. The existing application considers relatively low dimensional input data. While possibly an interesting result in itself, the authors may want to consider comparing their method to classical machine learning approaches (e.g. Random forests, SVMs, etc.) when using an analogous approach to train model parameters as used for the convolutional neural net in the application to rare cell types. I am concerned that the training procedure may have a similar or even larger effect than the choice of method.

Figure 1: I like panel a, visualizing the learnt representation. The authors may want to consider comparing the representation reconstructed when including a supervision signal during training versus employing a fully unsupervised training approach.

Figure 2: The compared results, either considering "filters achieving the highest area under the precision recall curve (AUPRC)" or results from an "automated procedure" are not sufficiently clear. The caption should more clearly describe what was done. How many filter were considered in total? How is the winner's curse avoided? See also major comments above.

Response to the referees' comments for manuscript NCOMMS-16-04501-T

This is the response to the reviewer comments for the paper entitled "Sensitive detection of rare disease-associated cell subsets via representation learning". This work presents CellCnn, a novel method for detecting cell subpopulations predictive of a given phenotype from single-cell measurements. The focus of our manuscript is on the detection of relevant, i.e. phenotype-associated cell populations that are **rare**, a non-trivial task that existing methods fail to solve.

We thank the reviewers for their overall positive evaluation on novelty and significance of our contribution, as well as for their constructive suggestions. We have identified and addressed the following main issues raised by the reviewers:

1. **Difficulty to interpret the outcome of the proposed procedure:** We added a more comprehensive and intuitive description as well as visualization of the phenotype-associated cell population identified by CellCnn. Furthermore, we discuss in greater detail situations and kind of cell populations identifiable by CellCnn.
2. **Potential overfitting of the proposed procedure:** We now consider an additional rare cell population detection scenario with clear definition of training, validation and test datasets that enables objective and fair comparison with baseline approaches and explicitly avoids overfitting.
3. **Additional comparison to further approaches:** We complemented benchmarks with all approaches suggested by the reviewers.

Please see below a detailed response to the comments of both reviewers. Reviewer comments are displayed in italic font, author responses follow in normal font.

Response to Reviewer 1:

While the topic of computational analysis of single cell data is likely of significant interest to a broad range of researchers, it may be difficult for biologists to interpret the findings of CellCnn. In particular, the filters do not necessarily correspond directly to cell populations and it would be useful to explore this more as this is both a strength of the algorithm and a potential pitfall for reading and interpreting the findings.

Indeed, the filters potentially select multiple cell types. To identify and further characterize such situations, we performed density-based clustering on the group of cells selected by a specific filter. For the CellCnn analysis of the stimulated PBMC samples, this analysis clearly defines multiple cell populations that exhibit distinct overall phenotypic profile. Such a situation is not

unexpected. CellCnn attempts to find a cell subset whose occurrence/frequency is associated with a phenotype. It is likely that such cell subset is characterized by a biological process that is active in various cell types, as for instance the Stat5 signaling response in both monocytes and dendritic cells upon GM-CSF stimulation. CellCnn learns a filter that selects cells from both cell types. The additional density-based clustering step on the selected phenotype-associated cells reveals its potential diversity with respect to e.g. cell type-defining surface markers. The results of this analysis are reported in the adapted **Figure 1d**, section “**Reconstruction of cell type specific signaling responses upon paracrine agent stimulation**”. The methodological details have been added to the Methods section in the paragraph entitled “**Characterization of identified cell subsets**”. Our response to the next reviewer comment elaborates further on the pitfalls of interpreting filter profiles and how to avoid these.

In Figure 1e, Filter 2 has both CD4 and CD8 as red for the filter (high) and Filter 1 has both CD4 and CD8 as blue (low). This is confusing. Are cells responding to Filter 2 double positive T cells (CD4+ CD8+ Ki67+)? Or are two different populations of cells able to respond to Filter 2 (CD4+ CD8- Ki67+ cells and CD4- CD8+ Ki67+ cells)? Or is this a typographical error?

We agree with the reviewer that these filters appear confusing. First, this pattern is not a typographical error. It seems contradictory to have one filter with positive weights for CD4 and CD8 while the other filter exhibits negative weights for the very same markers. The seeming contradiction is resolved by the opposite direction of the filter’s influence on the phenotype association (heatmap column labeled survival).

Second, it seems surprising to have positive filter weights for both CD4 and CD8. This result exemplifies what kind of statements can be done (or not be done) about the phenotype-associated cell population on the basis of the filter weights alone. In summary, we come to the conclusion that the filter weights alone are not useful for defining properties of the phenotype-associated cell populations. Instead, we suggest using the filters to select the phenotype-associated cell populations in the analyzed single cell data and report statistics such as histograms on the marker profiles of these selected populations.

The rationale for this interpretation procedure is the following. Neural networks can model very complex relationships between their input and output and, during training, their weights are optimized according to the training examples (here cell marker profiles) presented to them. The weights are not adequately trained for inputs (i.e. marker profiles) **never** presented to the network. This is a known issue for neural networks ¹. In the considered case, CellCnn was presented only to cells that were either positive for CD4 or CD8 and encoded the phenotype association of being positive for one of these markers by means of a single filter with weight on both CD4 and CD8. Not having been presented to a doubly positive CD4⁺CD8⁺ cell (only occurring during T cell development in the thymus and not in plasma samples) this filter choice results in equivalent predictions as having two separate filters, each with weight either CD4 or CD8. In summary, this issue does not compromise the predictive ability of the model for the effectively occurring marker profiles in the considered type of biological specimen. However this

issue again exemplifies that filter profiles should not be directly interpreted as marker profiles of phenotype associates cell subsets. In order to avoid any unnecessary confusion, we decided to remove the filter weight plots from the main manuscript and provide them as **Supplementary Figures** and instead report marker histograms of the selected cell populations for all applications (**Fig. 1e, 2c, 3e**) and color coded t-SNE projections (**Fig. 1c-d, 2b, 3a-d**).

Since the submission of the initial manuscript we have introduced an automated hyperparameter search procedure for the CellCnn network and report the new results on the HIV-cohort in the revised manuscript (see **Methods** sections “**CellCnn network architecture and training**” & “**Model selection and interpretation**” for details). It turns out that the heterogeneous cell population previously selected by filter 1 (naive T-cells) is now split into two populations (naive CD8+ T-cells and naive CD4+ T-cells) selected by two different filters. The phenotype-associated cell subset defined by these filters is equivalent to the cell subset defined by the previously reported filter.

How is it assessed whether the filters are over-fit to the training data? Would the filter in Figure 1c function effectively with only p-STAT5? How many other features in the filter are necessary?

Over-fitting is an important issue to take into account to avoid reporting spurious associations. CellCnn training is designed to avoid overfitting. Briefly, we always resort to a validation set of cell measurements, not used for training, to estimate the generalization error of a trained model and to compare across models with different hyperparameters (e.g. number of filters, learning rate). For a detailed description of the split of datasets, please see **Methods** section “**Datasets**” and for a description of the procedure taking advantage of these data splits, please see **Methods** sections “**CellCnn network architecture and training**” & “**Model selection and interpretation**”.

After achieving a CellCnn model that does not overfit, we investigate properties of the selected cell subsets. In the submitted manuscript, we provided the overlays of univariate marker distributions between the whole cell population and the specific cell population selected by the filter. For the revised manuscript, we added quantitative measures of the difference between these distributions. Difference between distributions is quantified via the Kolmogorov-Smirnov (KS) two-sample test statistic (see sections “**CellCnn overview**” and **Methods** section “**Characterization of identified cell subsets**”). Markers producing the highest KS test statistic are used to describe and interpret the selected cell populations. In the case of detecting PBMC cell subsets responding to GM-CSF stimulation (**Fig. 1c-e**) we find that the pStat5 distribution exhibits the greatest difference with the whole cell population. This information has been added to the section “**Reconstruction of cell type specific signaling responses upon paracrine agent stimulation**”. In particular, pStat5 would be sufficient for selecting the stimulated cells. While CellCnn analysis identifies correlated markers (Pearson product-moment correlation coefficient > 0.4) (**Fig. 1e**) that do not significantly improve the association with the stimulation condition, these markers recapitulate expected targets of GM-CSF stimulation.

It would be helpful to determine whether the computational approach is better than drawing simple cutoff gates, especially for the leukemia data in Figure 2.

We have added precision-recall curves for each single marker and its ability to enrich for leukemic blasts to address this comment (**Supplementary Fig. 6**). The computational approach (CellCnn) shows superior performance.

80% of the data was used to train the filters and 20% to test. How does it perform working with a smaller amount of training data (e.g. 33%) and the remaining data in the test setting?

We have re-analyzed the HIV cohort dataset using 50% and 33% of the samples as training data, instead of 67% that was used in our first analysis (**Supplementary Fig. 3**). The predictive performance on the new (now larger) test cohort remains competitive in terms of p-value of the log-rank test between the high- and low-risk groups. In addition, the two phenotype-associated cell populations identified when using a smaller amount of training data (50% or 33%) were also identified in our original analysis (with training data equal to 67% of the samples). In summary, we conclude that our results are not sensitive to the specific (and yet reasonable) split sizes for training and test data.

How different are the biological features used by CellCnn and Citrus to stratify in Figure 1f?

The three cell subsets identified by CellCnn (naive CD8+, naive CD4+, Ki67+ T-cells) coincide with the ones reported in the Citrus manuscript². When we applied Citrus on the same random training/test data partition as the one used for CellCnn, Citrus identified two out of these three cell populations (naive CD4+ and Ki67+ T-cells) plus one additional cell population exhibiting a CCR5+ CD28- CD8+ phenotype that was negatively associated with survival. The regression coefficient of the Ki67+ cluster (also negatively associated with survival) was one order of magnitude higher than the coefficient of the CCR5+ CD28- CD8+ cluster, denoting strong phenotype-association of the Ki67+ subpopulation (**Supplementary Fig. 4**). From these observations, we conclude that CellCnn and Citrus identify the same strongly survival-associated cell populations.

Would ensemble analysis be beneficial? Is either tool more likely to find a particular type of cell? Perhaps CellCnn is better at finding cells that do not form distinct subset clusters?

We believe that running both tools might be beneficial, because they follow a conceptually different approach. Detection of a cell population by both tools would be a strong indicator of this population's predictive power.

However, situations are conceivable where CellCnn correctly detects a cell population that Citrus does not detect. As the reviewer points out, CellCnn should be better at finding cells that do not form distinct subset clusters (in the space defined by all measured markers). We extended the section **Discussion** by these considerations and to illustrate this point, we have added a synthetic example (**Supplementary Fig. 8**). This example is constructed on the basis of the PBMC samples³ using the ten cell surface markers measured that clearly define distinct cell types. We assumed two different phenotypes A, B and introduced one additional marker that is predictive of a sample's phenotype. Samples from phenotype A only contain cells negative for the additional marker (generated from a gaussian distribution with mean=-1, std=1), whereas samples from phenotype B contain a subpopulation of cells that is positive for the additional marker (generated from a gaussian distribution with mean=1, std=1). This construction reflects the situation of having a cellular process (e.g. cell cycle) being active across various cell types for only one of the two considered conditions. The correct phenotype-associated cell subset is identified by CellCnn but cannot be identified by a cluster-based approach such as Citrus.

An important comparison would be to show that CellCnn can be trained on pre-treatment samples of leukemia and then still function well in real post-treatment samples to distinguish leukemia blasts / MRD from recovering healthy stem/progenitor cells.

We agree that this would be an important application of CellCnn. Previous work partially addresses this application scenario by resorting to conventional classifiers trained to detect a specific dominant leukemic clone determined by manual gating before treatment to monitor its potential expansion after treatment⁴. By construction, this approach is very sensitive in identifying the disease-associated clone before treatment, however any different clones that exhibit a selective advantage during treatment and expand during disease recurrence will remain undetected.

This situation motivates *de novo* detection of such novel clones by comparing cell populations changes before and after treatment, e.g. by CellCnn. However, as the reviewer points out, the comparison of post- versus pre-treatment samples could be confounded by recovering healthy stem/progenitor cells that are phenotypically distinguishable from their pre-treatment precursors, but not adversely associated with patient outcome. These observations reveal the inherent limitations of the post- versus pre-treatment comparison to exclusively identify leukemic blasts and distinguish them from recovering healthy stem/progenitor cells. While latter cells occur in all post-treatment samples, not all such samples comprise leukemic blasts. The hallmark of the samples comprising leukemic blasts is later relapse of the patient. Since information about patient relapse is not part of post- versus pre-treatment comparisons, it is not possible to unambiguously identify novel leukemic blasts from such comparisons.

We therefore propose a different type of comparison that would be informative and could be utilized by CellCnn to identify, novel leukemic blasts without being confounded by recovering healthy stem/progenitor cells. This comparison comprises only post-treatment samples of a cohort of patients with associated information about future relapse status (e.g. survival). CellCnn

could learn which cell subpopulations are associated with relapse status after treatment. These subpopulations will not correspond to the recovering healthy stem/progenitor cells since these are expected to be common to all samples regardless of the long-term treatment outcome. To the best of our knowledge, such a dataset for leukemia patients is not currently publicly available or in our hands and, because of this situation, we have not performed such an analysis here. However, this analysis is conceptually equivalent to the **detection of T cell subsets prognostic of AIDS-free survival** that is reported in the manuscript and successfully addressed by CellCnn.

It is not clear whether this approach simply provides a faster way to achieve things that are currently achieved with other tools.

We would like to stress that CellCnn enables us to identify **rare** cell populations at frequencies that, to the best of our knowledge, no other method achieves to detect. Specifically, we visualize and quantitatively demonstrate this unique capability in our revised AML benchmark by means of single cell precision-recall curves and newly added two-dimensional sample representations (see **Fig. 3 & 4**, **Methods** sections “**Model selection and interpretation**” and “**Computation of two-dimensional sample representations**” and below for detailed comments). Runtime considerations are only secondary.

We identify a couple of reasons why the existing methods considered fail to detect the rare cell types. These are discussed in detail in the sections **Main** and **Discussion**. Briefly, these are the following: for population-based approaches such as the moment-based classifier, the bulk signal masks the contribution from rare cell types. In approaches like Citrus, where clusters and corresponding cluster-based features are defined *a priori*, rare cell types might be either missed during clustering (e.g. merged in a bigger cluster and therefore the rare signal contribution is again masked) or, on the other extreme, if too many clusters are formed, the supervised predictor model suffers from too many uninformative features and decreased statistical power. CellCnn can be thought of performing a different type of clustering that is guided by the supervision signal and, therefore, is able to focus on cell types predictive of the considered phenotype, even if these cell types are extremely rare.

Organization and other points:

Figure 1 has three distinct parts: 1) the outline of the method, 2) the GM-CSF signaling data, and 3) the HIV data and comparison to Citrus. It may make sense to split these up, especially GM-CSF and HIV.

We have incorporated this suggestion in the revised version of the manuscript and report four instead of two figures. **Figure 1** comprises the outline of the method and demonstrates it on the basis of the GM-CSF signaling data. **Figure 2** exclusively presents the results on the HIV data. **Figure 3** demonstrates CellCnn’s ability to detect rare leukemic blast populations. **Figure 4**

summarizes the benchmark with a variety of baseline approaches that fail to detect these rare cell populations.

In Figure 1e, the features in the heatmap appear to be scaled to different scales for each feature.

We have addressed this comment by explicitly mentioning the scales of the learned filter weights throughout.

The manuscript may be challenging for biologist readers to follow in the current form as it uses field specific language. It would be stronger if a wider audience could read it, as machine learning tools for single cell data will be of general interest.

To address this comment, we have added summary sentences to give more intuition about the method and result interpretation. Additionally, we have introduced a **Glossary** for defining and explaining the field-specific terms used in the manuscript.

Response to Reviewer 2:

Major reported advantages are obtained in the regime of extremely rare cell types. I feel that the validation approach used for this task in particular is not sufficiently transparent and lacks rigor. Instead of using independent test sets to quantify model performance, the authors consider a validation set during training, in some instances directly to select filters, prior to reporting prediction accuracies on the same dataset. This approach, while I understand its motivation, makes it impossible to objectively compare and quantify the performance of alternative methods.

We agree with the reviewer's comment that an independent test set allows for objective comparison of the generalization performance of different methods. Ideally such a benchmark is done on a dataset from a large patient cohort, where rare cell types (e.g. therapy-resistant cell populations) would be expected to be associated with a particular phenotype (e.g. survival time). While such datasets exist for associations of abundant cell populations, as for instance with AIDS free survival and are already part of our study (**Fig. 2**), such a dataset for **rare** cell population associations is not currently in our hands or publicly available.

For the initial submission, we constructed such datasets for AML MRD synthetically, by computationally combining AML blast cells and cells from a healthy bone marrow sample into mixed synthetic samples on the basis of the data provided by Levine *et al.*⁵. In each case, the mixed sample was compared with a pool of control samples from four different healthy bone marrows. Each sample was initially split into a training (80%) and a validation (20%) set of cells. Model parameter fitting was performed on the training data. Performance evaluation on the validation data allowed for hyperparameter selection without overfitting. However, we agree with

the reviewer that an objective comparison with other baseline methods is not possible without having an additional test dataset not used for either parameter fitting or hyperparameter selection.

We revised this study to objectively compare CellCnn with existing methods with respect to detecting **rare** disease associated cell populations. Specifically, we assembled a benchmark dataset with clearly defined training/validation and test samples using data from healthy and AML bone marrow samples⁵. The AML samples are subdivided into two groups: group 1 comprises cytogenetically normal (CN) patients, whereas group 2 comprises patients with core binding factor translocation [t(8;21) or inv(16)] (CBF). We considered a three-class classification problem: given a new sample, we predict whether it belongs to the healthy, CN AML or CBF AML group. All methods are trained and possibly fine-tuned using the training/validation samples and their performance is assessed on the test samples. Test samples were constructed using cell measurements from patients other than those used for training/validation sample construction. For details see **Methods**, section “**Datasets**”. The results of this analysis, confirming that CellCnn achieves superior performance in the rare cell type setting, have been added to the revised manuscript (**Results**, section “**Detecting rare blast cell populations in leukemia**”, **Fig. 3 & 4**).

Furthermore, we evaluated CellCnn on a “personalized medicine” setting. This setting emulates the situation of monitoring a single patient in two different conditions (e.g. healthy and MRD) and asking which cell subpopulations are associated with these conditions for this specific patient. In contrast to the setting introduced above, this analysis does not aim at finding disease-associated cell subsets that generalize to a new cohort of patients (**Results** section “**Detecting rare blast cell populations in leukemia**”, **Supplementary Notes, Supplementary Fig. 7**).

While the “personalized medicine” setting is an interesting application for CellCnn, we emphasize that only the newly added benchmark study with clearly defined training/validation and test samples serves as valid proof of principle to demonstrate the enabling capability of CellCnn to detect disease-associated cell populations down to frequencies of 0.01 %.

The risk of overfitting when using deep models is large, simply because these models are highly parameterized.

While the risk of overfitting is still present, we would like to point out that, at its current form, CellCnn is not a deep neural network. It comprises only one hidden layer. As mentioned by the reviewer, mass cytometry measurements constitute relatively low-dimensional data and, for this reason, it was not necessary to add more hidden layers to achieve best possible and yet non-overfitted cell population detection performance.

Minor comments:

The authors may want to consider comparing their method to classical machine learning approaches (e.g. Random forests, SVMs, etc.) when using an analogous approach to train model parameters as used for the convolutional neural net in the application to rare cell types.

We have addressed this comment by comparing CellCnn to Random Forests and SVMs in the rare cell type setting (**Fig. 4**), and found all approaches to be inferior to CellCnn. Details can be found in the **Methods** section “**Baseline methods**”.

The authors may want to consider comparing the representation reconstructed when including a supervision signal during training versus employing a fully unsupervised training approach.

We followed this suggestion and used a denoising autoencoder as an example of a fully unsupervised training approach (**Methods** section “**Baseline methods**”). The representation learned by the autoencoder filters is very different as unsupervised training focuses on the task of reconstructing the majority of the cell abundance profiles and therefore ignores rare cell types. Consequently, this representation fails to dissect cell populations with respect to their phenotype label, such as disease state (**Fig. 4b**)

Figure 2: The compared results, either considering "filters achieving the highest area under the precision recall curve (AUPRC)" or results from an "automated procedure" are not sufficiently clear. The caption should more clearly describe what was done. How many filter were considered in total? How is the winner's curse avoided? See also major comments above.

Figure 2 from the initial submission has been split into two figures (now Fig. 3 & 4). Precision recall curves are reported in **Figure 4** for the new benchmark study with clearly defined training/validation and test samples (see reply to major comments above) and have all been generated with the same procedure described in **Methods** section “**Model selection and interpretation**”.

1. Nguyen, A., Yosinski, J. & Clune, J. Deep Neural Networks are Easily Fooled: High Confidence Predictions for Unrecognizable Images. *arXiv [cs.CV]* (2014).
2. Bruggner, R. V., Bodenmiller, B., Dill, D. L., Tibshirani, R. J. & Nolan, G. P. Automated identification of stratifying signatures in cellular subpopulations. *Proc. Natl. Acad. Sci. U. S. A.* **111**, E2770–7 (2014).
3. Bodenmiller, B. *et al.* Multiplexed mass cytometry profiling of cellular states perturbed by

small-molecule regulators. *Nat. Biotechnol.* **30**, 858–867 (2012).

4. Fišer, K. *et al.* Detection and monitoring of normal and leukemic cell populations with hierarchical clustering of flow cytometry data. *Cytometry A* **81**, 25–34 (2012).
5. Levine, J. H. *et al.* Data-Driven Phenotypic Dissection of AML Reveals Progenitor-like Cells that Correlate with Prognosis. *Cell* **162**, 184–197 (2015).

Reviewers' comments:

Reviewer #1 (Remarks to the Author):

The revised manuscript from Arvaniti and Claassen addresses reviewer comments well and is significantly improved. Overall, it is likely to be of high interest to researchers in computational cell biology and related fields. The explanations used in the rebuttal and edited text have helped to clarify how CellCnn can produce some biologically counter-intuitive results. Furthermore, the edits to the figures have clarified the main points of the manuscript and the inclusion of additional material in the supplement provides useful detail. The use of training and validation datasets and quantification of precision and recall help to assess the advances and highlight areas for future development. The updates to the discussion addressed the issues surrounding rare cell detection in leukemia and further ground the work in biological applications. The overall organization and readability of the figures, visualizations, and writing is improved and it is now easier to follow the authors' rationale in designing and testing CellCnn. One minor suggestion is that the Glossary of the key terms could be improved with additional biological context and references back to examples in the figures.

Reviewer #2 (Remarks to the Author):

The authors have substantially revised the paper and addressed the majority of my comments.

The paper and the present results are intriguing. The problem under consideration is certainly of interest and the application of convolutional neural networks as presented is innovative. However, I am still not entirely convinced of the practical utility of the method and have some remaining technical questions.

First, I have remaining technical concerns regarding the rare cell type classification. Reading through the methods implementation it appears that CellCnn uses a dedicated training procedure for rare labeled data ('biased towards outliers'). It is known that the performance of machine learning methods is affected by the balance of the training data. Is the performance of say the SVM model improved when composing the training data using a similar approach ?

Beyond technical challenges, it would be helpful to demonstrate superior performance unambiguously on real data in at least one use case. A particular strength of the model is its ability to learn rich representations but it feels like this is not fully exploited. Can the authors for example show that the model can be used to improve survival prediction when training on large amounts of additional multi-cell input without labels ? Similarly, is it possible to show improved cell prediction accuracy on non-synthetic data ? Fig. 4c suggests that benefits start to be visible starting from .1% frequency. So perhaps the cell classification on synthetically derived rare cell types can be avoided.

Specific comments:

Figure 4: The figure caption should better label which results are shown on empirical and using synthetically derived data.

Figure 4b: I don't think this comparison is very meaningful, except perhaps for the citrus-derived representations. PCA and the auto encoder do not use the labels and hence it is not surprising that the presentations are inferior. A better comparison would be to train the auto encode on a balanced datasets with equal proportions of all 3 classes.

Multi-cell inputs: Are there any modifications to the standard CNN such that the input cells are permutation invariant? A standard convolutional neural network will learn filters that exploit the order of the inputs. However, this does not makes sense for randomly sampled multi-cell inputs.

Supp. Fig. 9: Is the multi-input cell count (Supp. Fig. 9) a hyper parameter that is optimised on the validation set?

We are pleased that the reviewer comes to positive conclusions regarding the significance and originality of our work. We identify two remaining main concerns:

1. **Demonstration of detection of rare phenotype-associated cell populations for a real dataset.** We added a benchmark of CellCnn with a mass cytometry dataset acquired to study rare memory-like NK cells in the context of CMV infection. This benchmark demonstrates the capability of CellCnn to detect this infection-associated cell subset, in contrast to state-of-the-art baseline.
2. **Class label balance of reported datasets.** This concern is new to this revision round and was not part of the preceding concerns. We demonstrate that our training instances feature perfect class label balance, identify that this concern is related to a confusion of class labels defined for cell subsets with those for individual cells and address the concern by clarifying this distinction.

See below detailed responses to the specific reviewer comments. Reviewer comments are displayed in italic and responses in normal font.

First, I have remaining technical concerns regarding the rare cell type classification. Reading through the methods implementation it appears that CellCnn uses a dedicated training procedure for rare labeled data ('biased towards outliers'). It is known that the performance of machine learning methods is affected by the balance of the training data. Is the performance of say the SVM model improved when composing the training data using a similar approach ?

In our study, we use different procedures to create inputs (multi-cell inputs) for CellCnn from the single cell data. Each of these procedures subsamples the single cell data to generate multi-cell inputs. CellCnn associates **multi-cell inputs (instead of individual cells)** with a classification/ regression label. The label (e.g. survival, MRD/normal) of each multi-cell input coincides with the label of the cytometry sample which was used for subsampling. The balance of label frequencies in the training data is determined by the amount of multi-cell inputs generated from samples for each label class. For all classification tasks presented in this work, we considered only balanced training datasets, i.e. comprising an equal number of multi-cell inputs associated with each label. To highlight this point, we added the sentence "In all cases, we generated an equal number of multi-cell inputs associated with each label. " in Methods, section Cell subsets as multi-cell inputs.

The dedicated subsampling procedure that the reviewer refers to creates multi-cell inputs 'biased towards outliers' and is used in cases where extremely rare cell types are expected to be relevant for association with the sample phenotype (e.g. for the leukemia demonstrations). This procedure only affects the generation of the multi-cell inputs and has no influence on the label balance of the training dataset. In the presented results (Fig. 4), the Logistic Regression, SVM and Random Forest classifiers are trained on single-cell profiles subsampled using the same 'biased towards outliers' strategy that is used for creating the CellCnn multi-cell inputs. This is indicated in Methods, section Baselines.

Beyond technical challenges, it would be helpful to demonstrate superior performance unambiguously on real data in at least one use case. ... Fig. 4c suggests that benefits start to be visible starting from .1% frequency. So perhaps the cell classification on synthetically derived rare cell types can be avoided.

We agree with the reviewer that it is desirable to demonstrate CellCnn's superior capability to detect rare phenotype-associated cell populations for a real dataset. In this study, we report results on real datasets for phenotype association with abundant cell populations (Fig. 1:PBMC stimulation , Fig. 2: AIDS onset).

We complemented our study by an analysis of a single cell dataset where phenotype association manifests itself by means of a rare cell population. Specifically, we studied a recent mass cytometry based study of rare memory-like NK cells (frequency <1%) in the context of CMV infections (Horowitz et al. 2013). While CellCnn is able to detect this cell population from the ungated pool of peripheral blood mononuclear blood cells, state-of-the art cell population classification approaches like Citrus fail to identify the memory-like NK cell population. We have added a figure and adapted the main text to describe this benchmark.

The newly added NK cell benchmark complements the analysis of the realistic construction of MRD-like cases that aim to further investigate the putative scope of CellCnn's capability to detect extremely rare phenotype associated cell subsets. We argue that this investigation is informative since none of the data is simulated or relies on model assumptions of CellCnn (e.g. considering data sampled from a Gaussian mixture while assuming a Gaussian mixture model for clustering). All considered single cell profiles are mass cytometry measurements of healthy/leukemic bone marrow samples. The MRD-like samples are constructed by *in silico* spike-in of leukemic blasts into healthy bone marrow sample measurements. These constructed datasets do not constitute an unrealistically easy problem instance, since, in contrast to CellCnn, competing state-of-the-art methods fail to detect phenotype association of rare cell populations below 0.1 % frequency.

These results, besides the ones obtained from the analysis of the rare memory NK cell dataset, indicate the unique capability of CellCnn to detect rare phenotype associated cell populations in more real datasets to come.

A particular strength of the model is its ability to learn rich representations but it feels like this is not fully exploited. Can the authors for example show that the model can be used to improved survival prediction when training on large amounts of additional multi-cell input without labels ? Similarly, is it possible to show improve cell prediction accuracy on non-synthetic data ?

The reviewer suggests to extend the proposed supervised to a semi-supervised approach. This is an interesting extension of CellCnn. It is conceivable that in a situation with plenty of unlabeled data CellCnn could implement a deeper CNN architecture without overfitting and achieving richer representations to associate with phenotypes. However, such an extension constitutes a substantial conceptual as well as practical step that goes beyond the scope of this work. Furthermore, no suitable non-synthetic datasets are available to demonstrate the capabilities of such a semi-supervised approach. We therefore decided to stay with reporting the conventional supervised application setting of CellCnn, constructing sufficiently rich representations to detect phenotype associated cell populations an order of magnitude less frequent than state-of-the-art.

Specific comments:

Figure 4: The figure caption should better label which results are shown on empirical and using synthetically derived data.

We adapted the first sentence of the figure caption as follows: "Benchmark results on the identification of *in silico* spike-in rare leukemic blast populations for two acute myeloid leukemia (AML) subclasses (CN: cytogenetically normal, CBF: core binding factor translocation)."

Figure 4b: I don't think this comparison is very meaningful, except perhaps for the citrus-derived representations. PCA and the auto encoder do not use the labels and hence it is not surprising that the representations are inferior.

This comparison has been added to specifically address the request of the same reviewer from the first round of responses: "The authors may want to consider comparing the representation reconstructed when including a supervision signal during training versus employing a fully unsupervised training approach."

As expected, the considered unsupervised approaches (moment-based feature construction, auto-encoder) do not achieve a representation that separates the samples with respect to their class assignments. As pointed out by the reviewer and discussed in various sections of the manuscript, the reason for this failure can be attributed to unsupervised approaches not taking into account the class information while constructing a representation. While this result was expected, it is instructive to explicitly demonstrate to what extent application of unsupervised representation learning approaches fail to separate samples differing only in subtle cell-subset frequencies. In summary, we thank the reviewer to have suggested this analysis in the first place and conclude that the comparison with the chosen unsupervised approaches in Fig. 4b, in conjunction with the discussion of their inherent limitation to solve a supervised learning task, is worthwhile to be reported.

A better comparison would be to train the auto encoder on a balanced dataset with equal proportions of all 3 classes.

Throughout this study, we only consider analysis settings where we aim at associating **cell subsets** with labels (phenotypes) that characterize large heterogeneous groups of cells (cytometry samples). Our analysis **does not rely on a priori gated cell populations with label assignments for individual cells** (e.g. blast label for cells in a CD34+ gate). Consequently, we assume that labels for individual cells are unknown. CellCnn's purpose is to *ab initio* detect and define *a priori* unknown cell subsets (e.g. novel leukemic blasts) whose frequency associates with the label of the whole sample (e.g. relapse/no-relapse).

We understand the reviewer's suggestion to train an autoencoder on a balanced dataset with equal proportions of healthy cells, CN AML blasts and CBF AML blasts. To construct such a dataset with balanced cell counts for the relevant cell subsets, one would need to *a priori* define the relevant cell population gates. However, CellCnn is designed for association studies where exactly this knowledge is not available. We therefore believe that the suggested analysis does not help to demonstrate the capabilities of CellCnn.

Multi-cell inputs: Are there any modifications to the standard CNN such that the input cells are permutation invariant ? A standard convolutional neural network will learn filters that exploit the order of the inputs. However, this does not makes sense for randomly sampled multi-cell inputs.

Permutation invariance is achieved via the pooling operation (e.g. max or mean pooling), which is performed across **all cells** in a multi-cell input.

Supp. Fig. 9: Is the multi-input cell count (Supp. Fig. 9) a hyper parameter that is optimised on the validation set ?

In Supp. Fig. 9 we analyze the influence of two specific hyperparameters (number of cells per multi-cell input, number of multi-cell inputs generated per class) on CellCnn's ability to detect rare phenotype-associated cell subsets. For different settings of these two hyperparameters, we quantify the predictive performance of CellCnn as the area under the precision recall curve (AUPRC) evaluated on both the training and validation cells. We observed that the AUPRC values were very similar for the different settings, as long as the multi-cell inputs were sufficiently large (≥ 500 cells) to contain at least one phenotype-associated cell. Therefore, we concluded that CellCnn is not very sensitive to these two hyperparameters.

As pointed out by the reviewer, the two hyper-parameters (number of multi-cell inputs, number of cells per input) are optimized on the validation set. This information has been added in Methods, section Cell subsets as multi-cell inputs.

References

Horowitz, Amir, Dara M. Strauss-Albee, Michael Leipold, Jessica Kubo, Neda Nemat-Gorgani, Ozge C. Dogan, Cornelia L. Dekker, et al. 2013. "Genetic and Environmental Determinants of Human NK Cell Diversity Revealed by Mass Cytometry." *Science Translational Medicine* 5 (208): 208ra145.

REVIEWERS' COMMENTS:

Reviewer #2 (Remarks to the Author):

The authors have done good job in addressing my comments. I am still concerned that most importance advance is not clearly demonstrated using real datasets and instead relies on different flavours of synthetic benchmarks. Having said this, I have to admit that this is largely due to the lack of existing data at this point and the presented approach is certainly innovative and likely to more directly applicable in the future.

On balance, the paper makes an interesting contribution and hence I am supportive of publication. I would suggest to discuss these limitations of the current evaluation in the discussion section, to make these transparent to the reader.

We are pleased that the reviewers come to a final positive conclusion regarding the significance and originality of our work. See below detailed responses to the specific reviewer comments. Reviewer comments are displayed in italic and responses in normal font.

Reviewer #1 (Remarks to the Author):

The revised manuscript from Arvaniti and Claassen addresses reviewer comments well and is significantly improved. Overall, it is likely to be of high interest to researchers in computational cell biology and related fields. The explanations used in the rebuttal and edited text have helped to clarify how CellCnn can produce some biologically counter-intuitive results. Furthermore, the edits to the figures have clarified the main points of the manuscript and the inclusion of additional material in the supplement provides useful detail. The use of training and validation datasets and quantification of precision and recall help to assess the advances and highlight areas for future development. The updates to the discussion addressed the issues surrounding rare cell detection in leukemia and further ground the work in biological applications. The overall organization and readability of the figures, visualizations, and writing is improved and it is now easier to follow the authors' rationale in designing and testing CellCnn. One minor suggestion is that the Glossary of the key terms could be improved with additional biological context and references back to examples in the figures.

We have added references from the Glossary back to the main manuscript Figures, indicating where different supervised and unsupervised algorithms are used as well as where concepts as e.g. convolutional filters are illustrated.

Reviewer #2 (Remarks to the Author):

The authors have done good job in addressing my comments. I am still concerned that most importance advance is not clearly demonstrated using real datasets and instead relies on different flavours of synthetic benchmarks. Having said this, I have to admit that this is largely due to the lack of existing data at this point and the presented approach is certainly innovative and likely to more directly applicable in the future.

On balance, the paper makes an interesting contribution and hence I am supportive of publication. I would suggest to discuss these limitations of the current evaluation in the discussion section, to make these transparent to the reader.

We added the sentence "In this study, we have demonstrated the ability of CellCnn to *ab initio* identify phenotype-associated cell subsets in publicly available datasets with known ground truth." in the discussion section.